# Perceived Gap of Age-Friendliness among Community-Dwelling Older Adults: Findings from Malaysia, a Middle-Income Country

**DOI:** 10.3390/ijerph19127171

**Published:** 2022-06-11

**Authors:** Chee-Tao Chang, Xin-Jie Lim, Premaa Supramaniam, Chii-Chii Chew, Lay-Ming Ding, Philip Rajan

**Affiliations:** 1Clinical Research Centre, Hospital Raja Permaisuri Bainun, Ministry of Health, 30450 Ipoh, Perak, Malaysia; limxj.crc@gmail.com (X.-J.L.); premaa.rao@gmail.com (P.S.); chiichii.crcperak@gmail.com (C.-C.C.); prajan333@yahoo.com (P.R.); 2Perak State Health Department, Ministry of Health Malaysia, Jalan Koo Chong Kong, 30000 Ipoh, Perak, Malaysia; drding@moh.gov.my

**Keywords:** healthy aging, city planning, age-friendly city, community-dwelling, aging in place

## Abstract

Background: The United Nations predicts that the global population aged 65 years or above will double from 703 million in 2019 to 1.5 billion by 2050. In Malaysia, the older population has reached 2.4 million, accounting for nearly 8% of the population. This study aimed to evaluate the perceptions of the elderly on the importance and availability of the age-friendly features in eight domains specified by the Global Network of Age-Friendly City and Communities. Methods: This was a cross-sectional study conducted by structured face-to-face and or telephone interviews. Gap score analysis was performed for 32 items of the 8 age-friendly domains. The gap scores were categorized as follows: 0 = not important OR important and element available; 1 = important but unsure whether the element is available; 2 = important but element not available. The gap scores were then dichotomized into “yes” and “no”, and multivariate logistic regression analysis was subsequently performed. Results: From the 1061 respondents, the housing (55.4%) and transportation (50.7%) domains reported the highest mean perceived gap scores. Out of the 32 elements, the highest mean gap percent scores were observed in elderly priority parking bays (83.8%), home visits by healthcare professionals (78.9%), financial assistance for home modification and purchase (66.3%), and affordable housing options (63.6%). Respondents in the city center reported higher gap scores for modified restrooms, parks, volunteer activities, and the internet; respondents in the non-city center reported higher gap scores for nursing homes, healthcare professionals, and cultural celebrations. Age, location, marital status, income, duration of stay, physical exercise, internet access, and intention to continue working were found to be associated with a higher perceived gap in specific domains. Conclusion: The most significant unmet needs were detected in the housing, transportation, and employment opportunities domains. Considerable disparities in the perceived gap were detected between the older population in the city center and non-city center. To address shortcomings in the local age-friendly setting, coordinated municipal policies, political commitment, and benchmarking of existing age-friendly cities are warranted.

## 1. Introduction

The United Nations predicts that the global population aged 65 years or more will double from 703 million (9%) in 2019 to 1.5 billion (16%) by 2050 [1]. Malaysia is not exempted from the ageing population issue. The population aged 65 years and above reached 2.41 million in March 2021, accounting for approximately 8% of the Malaysian population [2]. The rate is on an increasing trend, and it is estimated to reach 20% of the population by 2056 [3].

In view of the rapid growth of aging population globally, the World Health Organization (WHO) took an initiative to improve the living environment for the elderly by launching the “Age-friendly Cities” program in 2005. According to the WHO initiative, an “age-friendly” city is one that encourages active aging; it maximizes possibilities for health, involvement, and security in order to improve people’s quality of life as they age [4]. In this program, a guide describes 8 main features of age-friendly living, namely housing, outdoor spaces, and buildings; transportation; job opportunities and civic participation; social participation; respect and social inclusion; communication and information; and community support and health services. It is published as an assessment tool for cities that have joined the WHO Global Network of Age-Friendly Cities and Communities (GNAFCC). Membership in the network does not imply age-friendliness. Rather, it represents cities’ dedication to listening to the needs of ageing populations, assessing and monitoring age-friendliness features, and collaborating across sectors and with older people to develop age-friendly physical and social settings, and the commitment to share one’s experience, accomplishments, and lessons learned with other cities and communities [5]. The establishment of an age-friendly city benefits not only the elderly but also all generations, whereby it enables citizens to be active and connected, which in turn generates growth in the economic, social, and cultural sector of a community [6].

Given the benefits of forming age-friendly cities and the fact that the population of Malaysia is aging, it is important for the Malaysian government to work toward formulating age-friendly cities. By far, in Malaysia, only Ipoh and Taiping city are members of GNAFCC. In particular, 16.7% out of 739,700 people in Ipoh city are more than 60 years of age [6]. While there are no preliminary data on Ipoh city that fulfils criteria for an age-friendly city, this study aims to evaluate the baseline characteristics of these 8 domains, namely housing, outdoor spaces and buildings, transportation, job opportunities and civic participation, social participation, respect and social inclusion, communication and information, and community support and health services in Ipoh city by exploring the perceptions of the elderly residing in Ipoh city on the importance, availability, and the perceived gap of the features specified under each domain to determine if Ipoh is an age-friendly city.

## 2. Materials and Methods

### 2.1. Study Design

This cross-sectional study was conducted from April to August 2021 by structured face-to-face or telephone interviews in the Ipoh City of Perak State, Malaysia. Cross-sectional study design was employed, as it is typical for population-based surveys, allowing estimation of prevalence and establishing association, which is appropriate for prompt public health planning, assessment and informing policies [7,8].

### 2.2. Study Population

We included older adults who are aged 60 and above, had lived in Ipoh for at least 6 months either continuously or intermittently, and could converse in Malay, English, Chinese, or Tamil. We excluded those who did not consent or were physically or mentally unfit to participate in the survey.

Participants were conveniently sampled at public hospitals, health clinics, recreational parks, and other public locations. The sample size was determined using a sample size calculator for estimation based on the formula to estimate a proportion with finite population correction and using a value of 0.5 (50%) as the desired proportion to be estimated [9]. Using an estimated total population aged ≥60 of 113,729 and a precision of 0.03, a total of 1068 samples were required for the survey. 

### 2.3. Study Instrument

We adapted the English version of the AARP Livable Communities—Great Places for All Ages Survey Questionnaire. It contained eight domains of a WHO age-friendly city: (D1) housing, (D2) outdoor spaces and buildings, (D3) transportation, (D4) health and wellness, (D5) social participation, (D6) volunteering and civic engagement, (D7) job opportunities, and (D8) communication and information.

Content validity and face validity were established through meetings with the state geriatrician, the Ipoh City Council and Perak State Health Department representatives. Content modifications were made to some of the items, such as types of homes, medical emergency response systems, and snow removal, in order to suit the local context. Questions on spoken languages, political views of participants, and season-related questions in the original questionnaire were removed.

Then, pre-tests were conducted to test the Malay, Chinese, and Tamil versions of the questionnaire involving representatives of both health and non-health staff. Minor amendments were made based on the suggestions and the English version of the questionnaire was finalized by the research team. The modified questionnaire underwent a translation process from English into Malay, Chinese, and Tamil following an internationally accepted translation standard [10,11,12]. The forward translation was carried out individually by one subject matter expert and one layperson, and the work was reconciled and finalized after a discussion session. It was then back-translated into English by two independent laypersons with a proficient command of English. Subsequently, we pre-tested the three sets of the questionnaire for each language among community elderly members. The questionnaire was further modified and finalized based on the findings. 

The finalized questionnaires contained two main sections (i) demographic characteristics (22 items) and (ii) availability and importance of the age-friendly components in the 8 domains (30 items). Participants rated the availability of the age-friendly elements on a 3-point scale: yes, no, not sure; the importance of each element on a 3-point scale: important, not important, not sure (Appendix A). 

### 2.4. Data Collection

Three pairs of data collectors were trained by the investigators in a one-day training session supplemented by a printed manual. Written or verbal consent was obtained from the subject before the interview began, for face-to-face and telephone interviews, respectively. Responses were recorded into the printed questionnaire by the data collectors and subsequently transcribed into the RedCap electronic data collection form. 

### 2.5. Data Analysis

The data, initially entered in RedCap, were exported and analyzed using the Statistical Package for Social Sciences (SPSS) version 20.0. The data were analyzed descriptively with frequencies and percentages (Appendix A), while gap score analysis was performed for 32 items of the 8 age-friendly domains. The gap scores were generated by comparing the importance and availability scores (10). The gap scores were categorized as follow: 0 = not important OR important and element available; 1 = important but unsure whether the element is available; 2 = important but element not available in Ipoh (Figure 1 and Appendix A). Responses with unsure importance or missing data were not included into the gap score analysis. The gap score was then dichotomized into “yes” and “no”. Subsequently, univariate binary logistic regression analysis was performed (Appendix A). Variables with *p*-values < 0.25 were included into the multivariate binary logistic regression model. Adjusted odds ratio (AOR) with 95% confidence interval were presented. 

The respondents were grouped into two sub-areas (city center and non-city center). Based on the Ipoh City Council mapping, Ipoh city was divided into five administrative zones. The only zone with a population of more than 10,000 was categorized as the city center, while the four remaining zones with a population of less than 10,000 were categorized as non-city center [13]. Demographic characteristics and gap scores of older people living in the city center and non-city center were further presented in sub-group analyses.

The individual item mean gap score was generated by dividing the total number of subjects with the perceived gap scores by the total number of valid responses for that item and multiplying them by 100. The domain gap scores were generated by summing the item mean gap scores and dividing by the total number of items in the particular domain. 

## 3. Results

### 3.1. Demographic Characteristics

A total of 1061 usable responses were included in the final analysis after excluding 7 incomplete questionnaires. Overall, the majority of respondents were comprised of those aged between 60 and 70 (677, 63.8%), females (556, 52.4%), with secondary education (517, 48.7%), living with at least one companion (990, 93.3%), declaring a monthly income of less than RM 2000 (931, 87.7%), staying in Ipoh for more than 60 years (503, 47.4%), and not working (865, 81.5%). About half of the respondents stayed in the city center (533, 50.2%), while the other half stayed in the non-city center (49.8%). The socio-demographic characteristics did not differ between the city center and non-city center respondents, except their age (*p* = 0.006) and health conditions (*p* = 0.001) (Table 1).

### 3.2. Perceived Gap Scores

Of the 8 domains, housing (55.4%, D1) and transportation (50.7%, D3) reported the highest mean perceived gap scores. Out of the 32 elements, the highest mean gap percent scores were observed in elderly priority parking bays (83.8%), home visits by healthcare professionals (78.9%), financial assistance for home modification and purchase (66.3%), and affordable housing options (63.6%) (Table 2).

The mean percent gap scores were compared between the city center and non-city center respondents. Significantly higher gap scores were observed among the city center respondents in 4 elements, including modified rest-rooms for people with disabilities (40.3% in the city center vs. 30.0% in non-city center, D2, *p* = 0.001); well-maintained parks and facilities (26.8% vs. 16.5%, D2, *p* < 0.001), a range of volunteer activities (29.9% vs. 22.2%, D6, *p* = 0.009) and free access to computers and the internet (56.0% vs. 49.0%, D8, *p* = 0.039). In contrast, respondents in the non-city center reported a significantly higher mean gap percent score in nursing homes for older people (28.3% in the non-city center vs. 16.4% in the city center, D5, *p* < 0.001), a variety of healthcare professionals (34.7% vs. 27.2%, D5, *p* = 0.009) and the variety of cultural celebrations (35.9% vs. 24.0%, D5, *p* < 0.001) (Table 2).

### 3.3. Multivariate Binary Logistic Regressions

For the housing domain (D1), those who were aged 81 and above had significantly lower odds of a perceived gap (OR: 0.54; CI: 0.31–0.93), while those who actively exercised (5–7 times weekly) (OR: 1.53, CI: 1.01–2.33) and had internet access (OR: 1.50, CI: 1.10–2.04) tended to have a higher perceived gap. In terms of outdoor spaces/buildings (D2) and transportation (D3), those who have lived in Ipoh for more than 60 years have a higher perceived gap. Participants who lived in Ipoh between 11 and 30 years (OR: 2.89, CI: 1.30–6.40) and more than 60 years (OR: 2.34, CI: 1.15–4.77) reported a higher perceived gap in health and wellness (D4).

In the aspect of social participation (D5), those living outside of the city center (OR: 1.38, CI: 1.07–1.79), earning more than RM 4800 per month (OR: 2.98, CI: 1.11–8.01) and having access to the internet (OR: 1.36, CI: 1.04–1.77) were associated with higher odds of perceived gaps. As for the volunteering and civic engagement domain (D6), those in the non-city center who have lived in Ipoh for more than 60 years (OR: 0.71, CI: 0.53–0.94) and with no intention of continuing working (OR: 0.69, CI: 0.49–0.95) reported a lower gap score, while those with access to the internet reported a higher gap score (OR: 1.64, CI: 1.23–2.19). Those with moderate (OR: 1.68, CI: 1.06–2.67) and higher incomes(OR: 3.20, CI: 1.30–7.84), as well as those who have access to the internet (OR: 1.50, CI: 1.14–1.98), have a larger perceived gap in the domain of job opportunities (D7), while those who exercise regularly (OR: 0.67, CI: 0.46–0.99) have lower expectations in this respect. In terms of community information (D8), a higher gap score was seen in those unmarried (OR: 1.66, CI: 1.13–2.45) and those with access to the internet (OR: 1.39, CI: 1.04–1.85) (Figure 2).

## 4. Discussion

Out of eight domains, housing recorded the highest perceived gap, with a substantial gap existing in affordable housing options and financial assistance for either house renovation or purchase. According to the WHO, housing contributes to a range of positive health outcomes, particularly in the older population [14]. Housing affordability is a major concern for older people, especially vulnerable individuals with low incomes [15,16]. In European countries, the provision of housing subsidies was proposed to relieve the impact of housing costs on aging in place, which restricts one’s capacity to pay for other essentials [17]. Meanwhile, simple housing modifications such as installing lighting and grab bars in the bathrooms have been found to improve daily activity performance and mental health [18]. Locally, there are limited policies to protect the rights of senior citizens in housing options. Moreover, both public and private housing developers rarely provide affordable housing options for the aged [19]. Additionally, the concepts for older individuals such as co-housing, group living, or reconstruction of deserted vacant houses that may be instrumental for the aging population are yet to be popularised in this country [20]. The current housing financial support program by the government, such as the My First Home Scheme, could be expanded to cover housing modifications and house purchases for older adults [21].

We observed a notable need in transportation in terms of accessibility, cost, and availability of public transport to key destinations for older individuals, which is incongruent with developed European nations where it is evident that older adults are particularly vulnerable to transportation barriers [22]. Although the Malaysian government offered senior citizens a public transportation discount [23], the gap in the basic features of transportation remained significant. This may be related to the issues associated with health and mobility limitations [24]. In addition, public transportation in Ipoh city relies on a fixed route system, where passengers must travel along designated routes with predetermined schedules and designated pickup and drop-off stations [25]. Ergonomic improvement of public transport, addressing the physical challenges associated with boarding and alighting buses or terminals, covering the distance between stops and houses, may encourage older individuals to use public transportation [26]. Flexible transport services are gradually integrated into the UK, European, and American public transport systems to complement the traditional routes covered by small buses, minibuses, or maxi-taxis [27]. Flexible transport services employed interactive booking and reservation systems which could dynamically assign passengers to vacant vehicles and optimizes the routes [28].

Parking difficulties among the elderly are similar to those faced in other countries irrespective of socio-economic development. That is, the need for priority parking bays for senior citizens in close proximity to destinations, drop-off, and pick-up bays, with adequate space for them to get in and out of a car without obstruction, has been raised. Dedicated cycling and walking lanes distanced from car lanes [20], supplemented with audio or visual pedestrian crossings, should be considered to enhance the walkability and age-friendliness of the neighbourhood in future town-planning.

It is unsurprising that our respondents reported a relatively lower overall perceived gap in the healthcare domain, as there is a large network of government-funded primary care clinics and hospitals [29]. However, in congruence to previous findings, older individuals in non-city centers faced greater difficulties in accessing healthcare compared to their city center counterparts, and both reported remarkable perceived gaps in terms of home visit service [30]. The growing number of disabled older people with comorbidities necessitates the expansion of home health care services, which have been shown to reduce mortality and hospitalisation [30,31]. For instance, the Japanese government has promoted physician-led home-visit care for frail and disabled people [31] and such services, yet to be expanded to geriatric patients in this country, can be first considered for implementation in an ageing city like Ipoh.

The gap score was reported to be significantly higher in specialized care and nursing homes for the aged who reside in non-city center areas than those in the city center. The majority of specialist facilities needed by the elderly are placed in the city center [32,33]. Increased funding for geriatric specialist training may be necessary to meet the needs of older adults, particularly those who live outside of city centers. Despite the Malaysian Department of Social Welfare offering long-term care residences for dependent people who have no family support [29], nursing homes in Malaysia are still in short supply [34]. This disparity in nursing home availability and distribution, as suggested by this study, warrants the urgent attention of the welfare department to tackle this problem by providing incentives to nursing homes set up in non-city centers in Ipoh.

Senior citizens living in Ipoh city centers perceive greater barriers to accessing public restrooms and well-maintained parks as compared to their non-city center counterparts. Outdoor space has a significant effect on older adults’ mobility, independence, and emotional and psychological well-being, all of which are factors that influence the quality of life [35,36]. The use of outdoor spaces was strongly influenced by aesthetic features, practical components such as restrooms, and park maintenance [35]. Clean, safe, and accessible public toilets are important for older adults, particularly those with incontinence. Improving the accessibility and safety of parks, for instance, using wide and flat-surfaced pavements separated from cyclists, may promote physical activity among older adults [20]. To preserve outdoor spaces, a multifaceted approach should be taken, including increased budget allocation, integration of age-friendly outdoor features into local city council town-planning policy, collaboration among public and private stakeholders, and public awareness campaigns [37].

Two-fifths of our respondents reported perceived gaps in job opportunities, and this was more likely among seniors with moderate to higher incomes after controlling for demographic characteristics. Employment is crucial for the elderly’s financial well-being since it offers both a source of income and, in certain cases, benefits such as pensions and health insurance [38]. A Dutch study revealed that managers were unlikely to re-employ older employees after mandatory retirement, and re-employment of those who were willing to accept a lower salary was favoured [39]. Measures should be taken to reduce discrimination against older people, re-training and upgrading their skill sets, providing special working arrangements and revising the legal framework to increase older workers’ employment opportunities [40,41].

The lack of employment prospects may result in financial and economic volatility. While the state has the largest resources, a paradigm shift is difficult, as administrators are geared towards bureaucracy rather than implementing new policies. Hence, the government should take precautionary actions in the realm of social protection through revision of the tax system and tapping into existing private capacity to complement current public resources [42]. Meanwhile, individuals’ demands for social safety, self-protection skills and resilience should be understood and addressed through social inclusion policies. Better protection of the most vulnerable allows society to maximize existing capacities without duplicating efforts, resulting in a better alleviation of social anxiety. Recognising vulnerabilities produces synergies between state social protection and social policies, leading to innovative interventions and reorienting social protection [43].

Gap scores were relatively lower in social participation as compared to other domains. Yet, it is an important domain in developing community vitality, promoting physical and mental health, avoiding disabilities [44], and reducing the risk of death and dementia [45]. The importance of social participation should be promoted to older adults. The lack of variety in cultural celebrations and social clubs for hobbies was significantly raised by the non-city center respondents as compared to those in the city center. Social activities that are primarily organized in the city center pose considerable challenges for the participation of elderly who live in areas outside of the city center. A study shows that the main factors driving participation in social activity in older adults are easy access and being informed about transportation options [44]. Most social events could be fixed at the city center due to the consideration of available facilities. The organizers could always make an effort by arranging public transport service for cultural events, and providing priority parking and drop-off bays for senior citizens, and ensure that the information is clearly stated in the promotion advertisement. A senior center that organizes health programs and volunteering activities such as clubs for hobbies, educational courses, and exercise classes is suggested as a measure to increase social interaction among senior citizens [45]. The currently available senior center concept in Ipoh is limited and mainly located at the city center [46,47]. Nevertheless, prior to implementing any interventions prioritizing the needs of senior citizens in non-city center areas, in-depth studies are needed to determine the proximity of the elderly to the facilities, caregivers’ support, transportation, neighborhood security, and user-friendliness of the walking environment [48].

Respondents who have been staying for more than 60 years in non-city centers with no intention of continuing working, perceive a lesser need for civic engagement. In spite of the lesser interest in this domain, engagement in civic activities is proven to reduce mortality rates and encourage the elderly to stay healthy as it is associated with making oneself feel useful and responsible to others, continuing working on personal growth and development instead of receiving instrumental help [49,50]. The benefits of taking part in volunteering activities should be actively promoted to older adults while at the same time adequately addressing the aspects of accessibility, expectations, information, incentives, and facilitation, that in turn would foster the participation of the aged population in civic activities [51].

A perceived gap in communication and information was identified among those who were unmarried and had internet access. The use of the internet for communication could help to reduce social isolation, loneliness, and depression, and enhance social support in the elderly [52]. Even though respondents have access to the internet, which can be used for communication and information, their needs for this domain remained unfulfilled, suggesting that other aspects such as technology infrastructure, internet speed and coverage, digital divide, or a preference for face-to-face communication could be further researched. Another possible reason for this observation could be attributed to the greater need of single people for internet assessment to achieve a higher level of social capital [53]. Evidence shows that being single or a frequent internet user in older populations is associated with higher levels of social capital, and this is important to maintain good health and wellbeing [54].

Roughly half of the respondents, disregarding their geographical location, expressed a significant need for free device and internet access. Unaffordability of devices and internet access among the elderly is not uncommon, and this problem happens even in developed countries, such as the UK. Inequality in the digital divide, and the distribution of technological infrastructure reduce internet access for residents living in different locations, affecting all age groups and resulting in digital poverty [20,55]. A slightly lower gap score rated in the need for free devices and internet access by respondents in the non-city center could be explained by the continuous effort of the Malaysian government to increase internet access for residents in the non-city center area. This has been reflected in allocating funds in the eighth and ninth Malaysia plans and in setting up telecenters in the rural community by establishing facilities for the internet and computer training rooms for use by the rural community [56].

### Strength and Limitations

To the best of our knowledge, this is the first study in the Malaysian context that evaluates a city’s age friendliness. This study surveyed the population of older adults that were distributed equally in the city and outside the city area in Ipoh. The findings served as a baseline input for local city councils to make improvements to Ipoh city not only to meet the benchmark of GNAFCC but also to plan for short- and long-term intervention in creating an age-friendly environment. In the global context, the information from this study adds knowledge to existing aging research that allows researchers to make comparisons among the age-friendly cities. Studies with comparable methodology may be replicated in other countries to validate the findings in future.

Convenient and snowball sampling methods may not encompass all social-economic groups of older populations in Ipoh. Sampling was not stratified based on city center and non-city center, thus sample size may not be adequately powered for the subgroup analyses. Nonetheless, the study analysis shows that the distributions of demographic characteristics were not significantly different between those who stayed in the city center and those who did not. The differences in the need for an age-friendly between the two groups can thus be compared, and this allows policymakers to identify and address the inequalities experienced by the aged population, including those residing in the city center or non-city center of Ipoh.

## 5. Conclusions

Out of the eight domains, the most notable unmet needs were observed in the housing, transportation, and job opportunities domains. Distinct differences in perceived gap were also observed between older populations in the city center and non-city center in six out of the eight domains. A multifaceted approach is recommended to specifically address the unmet needs among older adults in the local context. Public-private housing partnership model could be introduced to revolutionize housing policy and financing assistance scheme to improve house purchase and modifications affordability. Expansion of home visit services is essential to maintain continuity of care for patients who have been discharged, support family members and reduce readmission. Continuation of public transport fare subsidy policy may encourage its usage among older adults. Provision of flexible working arrangements and revision of legislative framework may reduce discrimination and increase job opportunities for older people. On a broader scale, concerted municipal strategies, political commitment, and benchmarking of established age-friendly cities are warranted to address specific gaps in the age-friendly city context.

## Figures and Tables

**Figure 1 ijerph-19-07171-f001:**
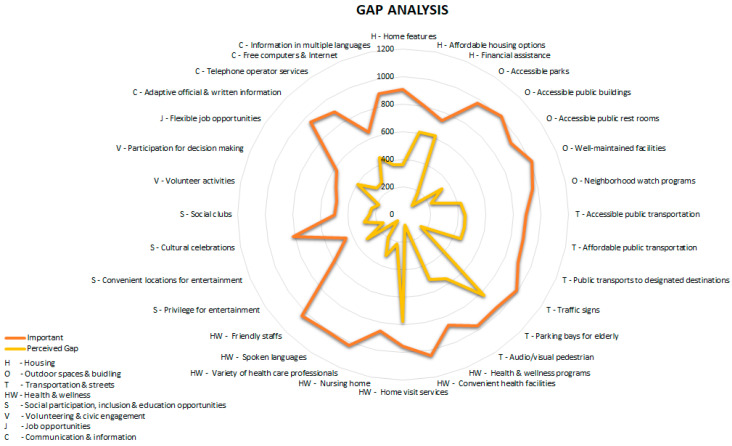
Distribution of the perceived gap in the elements of eight age-friendly domains.

**Figure 2 ijerph-19-07171-f002:**
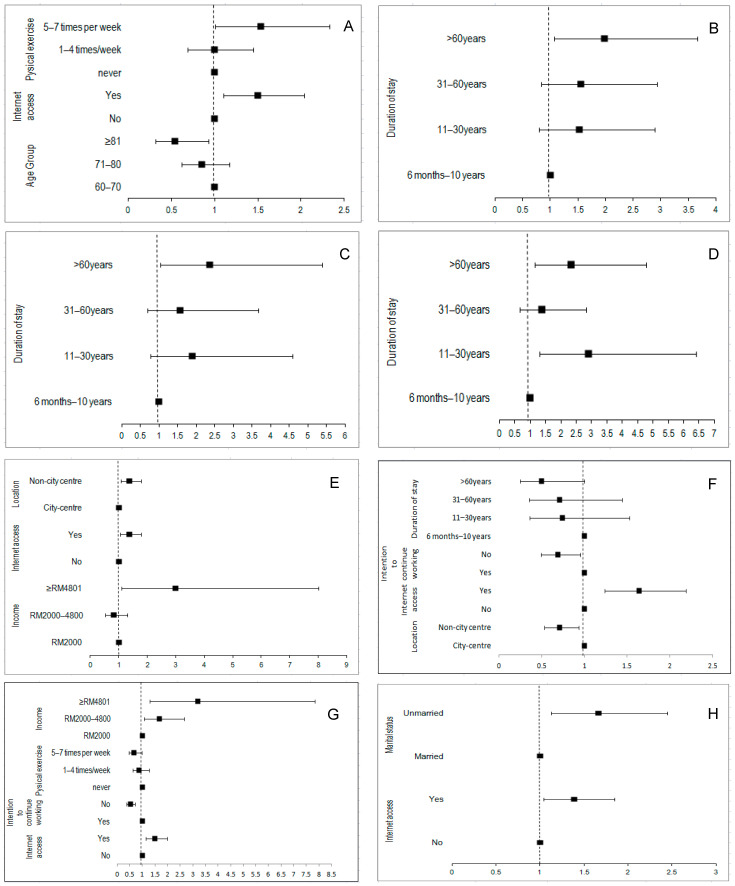
Multivariate binary logistic regressions assessing significant factors associated with presence of perceived gap on the eight age-friendly domains. (**A**) Housing, (**B**) outdoor spaces and buildings, (**C**) transportation, (**D**) health and wellness, (**E**) social participation, (**F**) volunteering and civic engagement, (**G**) job opportunities, (**H**) community and information.

**Table 1 ijerph-19-07171-t001:** Socio-demographic characteristics of the respondents in city center and non-city center of Ipoh (*n* = 1061).

	Overall (*n* = 1061)		City Center (*n* = 533)		Non-City Center (*n* = 528)		*p-*Value
	*n*	%	*n*	%	*n*	%	
Age (years) (Mean, SD)	68.69, 6.58		69.24, 6.92		68.13, 6.17		**0.006**
60–70	677	63.8	323	60.6	354	67.0	0.042
71–80	314	29.6	167	31.3	147	27.8	
81 and above	70	6.6	43	8.1	27	5.1	
Gender							
Male	505	47.6	254	47.7	251	47.5	0.970
Female	556	52.4	279	52.3	277	52.5	
Marital status							
Married	875	82.5	442	82.9	433	82.0	0.694
Unmarried	186	17.5	91	17.1	95	18.0	
Ethnicity							
Malay	381	35.9	198	37.1	183	34.7	0.296
Chinese	263	24.8	140	26.3	123	23.3	
Indian	402	37.9	189	35.5	213	40.3	
Others	15	1.4	6	1.1	9	1.7	
Education							
No formal education	51	4.8	27	5.1	24	4.5	0.151
Primary	356	33.6	191	35.8	165	31.2	
Secondary	517	48.7	241	45.2	276	52.3	
Tertiary	137	12.9	74	13.9	63	11.9	
Living status							
Living alone	71	6.7	40	7.5	31	5.9	0.287
Not living alone	990	93.3	493	92.5	497	94.1	
Income, RM (Mean, SD)	1221.1, 1488.1		1190.8, 1544.1		1251.6, 1430.2		0.507
Less than RM 2000	931	87.7	468	87.8	463	87.7	0.818
RM2001-RM 4800	101	9.5	49	9.2	52	9.8	
More than RM4801	29	2.7	16	3.0	13	2.5	
Perceived opinion Ipoh as a place for senior citizen to live							
Good	941	88.7	468	87.8	473	89.6	
Moderate	117	11.0	64	12.0	53	10.0	0.504
Poor	3	0.3	1	0.2	2	0.4	
Duration of stay in Ipoh (years)							
6 months–10 years	47	4.4	26	4.9	21	4.0	0.722
11–30 years	210	19.8	104	19.5	106	20.1	
31–60 years	301	28.4	157	29.5	144	27.3	
More than 60 years	503	47.4	246	46.2	257	48.7	
Health condition							
Healthy	224	21.1	90	16.9	134	25.4	**0.001**
Active but with underlying diseases	785	74.0	409	76.7	376	71.2	
Inactive/with restricted mobility	52	4.9	34	6.4	18	3.4	
Possess health care coverage/insurance							
Yes	401	37.8	199	37.3	202	38.3	0.757
No or unsure	660	62.2	334	62.7	326	61.7	
Engagement of physical exercise in a week							
Never	196	18.5	95	17.8	101	19.1	0.598
Sometimes (1–4 times)	502	47.3	248	46.5	254	48.1	
Frequently (5–7 times)	363	34.2	190	35.6	173	32.8	
Access the Internet							
Yes	521	49.1	262	49.2	259	49.1	0.973
No	540	50.9	271	50.8	269	50.9	
Current employment status							
Employed	196	18.5	104	19.5	92	17.4	0.381
Unemployed	865	81.5	429	80.5	436	82.6	
Intention to continue to work for as long as possible							
Yes	230	21.7	120	22.5	110	20.8	0.506
No	831	78.3	413	77.5	418	79.2	

**Table 2 ijerph-19-07171-t002:** Mean gap percent scores of the WHO AFC items and domains in city center and non-city center.

			**City** **Center**	**Non-City Center**	**Difference**	**Overall**	** *p* ** **-Value**
**Domain**	**Elements**	**Valid *n***	**Mean Gap %**	**Mean Gap %**	**Mean Gap %**	**Mean Gap %**	
Housing (D1)	Equipped with home safety features	990	33.6	39.3	5.7	36.4	0.064
	Affordable housing options	958	64.8	62.3	2.5	63.6	0.424
	Financial assistance for home modification and purchasing	932	66.5	66.2	0.3	66.3	0.927
	Overall mean gap percent score		54.9	55.9	1.0	55.4	0.334
Outdoor spaces and buildings (D2)	Accessible parks and recreational areas	992	19.3	17.1	2.2	18.2	0.374
	Accessible public building and facilities	1022	10.0	8.1	1.9	9.1	0.290
	Rest rooms accessible to people with physical disabilities	963	**40.3**	**30.0**	**10.3**	**35.1**	**0.001**
	Well-maintained parks and facilities	1022	**26.8**	**16.5**	**10.3**	**21.7**	**<0.001**
	Neighborhood watch program	989	44.4	42.2	2.2	43.3	0.479
	Overall mean gap percent score		28.2	22.8	5.4	25.5	0.060
Transportation and streets (D3)	Accessible public transportation	919	48.6	48.9	0.3	48.7	0.931
	Affordable public transportation	911	48.6	51.3	2.7	49.9	0.408
	Public transport travel to key destinations	921	50.6	47.3	3.3	49.0	0.305
	Easy to read traffic signs	1001	17.0	14.6	2.4	15.8	0.305
	Priority parking bays for elderly	986	83.5	84.0	0.5	83.8	0.841
	Audio/visual pedestrian crossings	982	57.1	57.1	0.0	57.1	0.993
	Overall mean gap percent score		50.9	50.5	0.2	50.7	0.906
Health and wellness (D4)	Health and wellness programs	927	53.6	55.9	2.3	54.8	0.483
	Conveniently located health facilities	1044	8.1	6.4	1.7	7.3	0.277
	Home visit by healthcare professionals	978	78.0	79.9	1.9	78.9	0.453
	Nursing home for older people	971	**16.4**	**28.3**	**11.9**	**22.3**	**<0.001**
	A variety of healthcare professionals including specialists	1030	**27.2**	**34.7**	**7.5**	**30.9**	**0.009**
	Health care professionals who speak different languages	1033	18.6	19.4	0.8	19.0	0.746
	Respectful and helpful heath care staff	1039	5.3	6.0	0.7	5.7	0.638
	Overall mean gap percent score		29.6	32.9	3.3	31.3	0.051
Social participation, inclusion and education opportunities (D5)	Privilege for entertainment	902	34.5	35.2	0.7	34.8	0.827
	Convenient location for entertainment	862	17.2	19.6	2.4	18.4	0.360
	A variety of cultural celebration	955	**24.0**	**35.9**	**11.9**	**29.8**	< **0.001**
	Social clubs for hobbies	872	**31.5**	**25.2**	**6.3**	**28.3**	**0.038**
	Overall mean gap percent score		**26.8**	**28.9**	**2.1**	**27.9**	**0.014**
Volunteering and civic engagement (D6)	A range of volunteer activities	900	**29.9**	**22.2**	**7.7**	**26.1**	**0.009**
	Opportunity to participate in decision making bodies	905	23.4	19.3	4.1	21.3	0.125
	Overall mean gap percent score		**26.7**	**20.8**	**5.9**	**23.7**	**0.007**
Job opportunities (D7)	Flexible job opportunities	968	41.3	40.3	1.0	40.8	0.770
	Overall mean gap percent score		41.3	40.3	1.0	40.8	0.770
Community and information (D8)	Readable written information	987	**30.5**	**24.8**	**5.7**	**27.7**	**0.047**
	Telephone operator services adapted to the needs of seniors	919	32.7	28.3	4.4	30.5	0.144
	Free access to computers and internet	861	**56.0**	**49.0**	**7.0**	**52.4**	**0.039**
	Information available in different languages	963	36.2	40.3	4.1	38.2	0.197
	Overall mean gap percent score		38.9	35.6	3.3	37.2	0.307

Note: *p*-value generated using Chi-square.

## Data Availability

The dataset used in this study can be obtained from the authors upon reasonable requests.

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
