# Peer review of "Perceived Gap of Age-Friendliness among Community-Dwelling Older Adults: Findings from Malaysia, a Middle-Income Country"

_ijerph, 2022, doi:10.3390/ijerph19127171_

Round 1
Reviewer 1 Report
Manuscript ID: ijerph-1694119
Title: Perceived gap of age-friendliness among community-dwelling older adults: Findings from a middle-income country
Authors: Chee Tao Chang, Xin Jie Lim, Premaa Supramaniam, Chii Chii Chew, Lay Ming Ding, Philip Rajan
Comments and Suggestions for Authors:
The topic and the idea of the study is very valuable and corresponds very well to the section of the IJERPH journal. However many elements of the manuscript should be improved/present in better form to highlight this important topic and especially interesting results obtained by the Authors.
1. The Title is in line with the presented study, it explain the studied context. However, the Malaysia should be added to the title.
2. The Abstract is very long, however acceptable - it presents very clear the research. However, the first sentence of background should be directly related to Malaysia - there is no information in Abstract that the study concerns this country!
The keywords are related to the topic.
3. The Introduction is not much developed, most information is related to the well known general information. Especially the main 8 domains, which are used by Authors, are not identified and argued - it is not clear why they are so important for older adults in the context of benefits and age-friendly cities. This is a weakness of the manuscript and must be improved/completed. Without those information the background is uncompleted, and thus the direction of research is not clear. Also the number of sources included in Introduction is insufficient as a scientific background for so deep and developed study presented in the manuscript.
4. The aim of the study is short and rather general. Regarding the above-mentioned 8 domains, they should be repeated there to be clear if all of them are studied. There are no hypotheses formulated. This part must be developed/ improved.
5. Generally, the presentation of section Material and Methods should be divided into subsections to present clearly the Material, e.g. sample selection and number of respondents divided into age, how many of them represent the city centre and non-city centre of Ipoh, also the form / scope of questionnaire should be described – due to many important aspects asked, they should be presented in a table, etc.
The used Methods are not clear – they need to be explained in a more scientific way, more professional, also they must be clearly referred to the sources to confirmed them as typical or not for that kind of study and argued. The present form of this section is difficult to understand, especially the stages of the study and their role for possible results.
6. In the section of Results, the characteristics of respondents could be a bit developed to highlights more its main specific in the context of place of living or status; the direction in this case should follow then the order of presentation of differences in the further presentation of results.
The description of results itself is not much developed. The division into 8 domains is correct, however the high number of diversity between different types of respondents make it a bit difficult to notice.
The organization/presentation of Table 3 is weak, it must be improved, e.g. divided into separate or some more individual sections, to present the completed data better.
There is a lack of description/presentation in the text related to the significance of obtained results, for example in the relation to demographic characteristics of respondents, etc., presented in Table 3. That aspects also needs interpretation.
7. Despite some gaps in most sections, Discussion is quite strong part of the manuscript. However, the description is too long and due to the presentation of many (too many?) aspects in a very descriptive way, it becomes distracting and it from is a kind of a long “story” - it is difficult to find out what are the most valuable results of the study? The Authors should present the discussed aspects more synthetically and focus on their specifics /innovation - to present clearly the basic gaps of age-friendliness among community-dwelling older adults, and relate them directly to other studies.
8. Conclusions are very short, and rather repeat general information from the Discussion, than summarize the manuscript and findings.
It must be clearly defined what the conducted research brings, especially in scientific (and practical?) context. Also the main direction in which interventions should be taken in all 8 domains must be proposed. In its present form, the conclusions are insufficient and not much linked to the research findings and their discussion.
9. Others:
- Figure 1 has a poor resolution and is not legible - it needs improvement.
- there are some punctuation errors in the text - lack of dots, etc.
- the way of writing references in text should be in square bracket
Regarding the above-mentioned weaknesses I can not recommend this manuscript to be published in its present form. It needs major revision in my opinion.
Author Response
- The Title is in line with the presented study, it explain the studied context. However, the Malaysia should be added to the title.
Author’s reply: Malaysia was added in the title
- The Abstract is very long, however acceptable - it presents very clear the research. However, the first sentence of background should be directly related to Malaysia - there is no information in Abstract that the study concerns this country!
Author’s reply: Added in abstract: In Malaysia, the older population has reached 2.4 million, accounting for nearly 8% of the population.
The keywords are related to the topic.
- The Introduction is not much developed, most information is related to the well known general information. Especially the main 8 domains, which are used by Authors, are not identified and argued - it is not clear why they are so important for older adults in the context of benefits and age-friendly cities. This is a weakness of the manuscript and must be improved/completed. Without those information the background is uncompleted, and thus the direction of research is not clear. Also the number of sources included in Introduction is insufficient as a scientific background for so deep and developed study presented in the manuscript.
Author’s reply:The introduction was revised with additional context on the benefits of age-friendly cities and the meaning of it, with additional references as support: According to the WHO initiative, an "age-friendly" city is one that encourages active aging; it maximises possibilities for health, involvement, and security in order to improve people's quality of life as they age (Plouffe et al. 2010). Membership in the Network does not imply age-friendliness. Rather, it represents cities' dedication to listening the needs of ageing populations, assessing and monitoring age-friendliness features, and collaborating with older people and across sectors to develop age-friendly physical and social settings, and the commitment to share one's experience, accomplishments, and lessons learned with other cities and communities (Flores et al. 2019).
- The aim of the study is short and rather general. Regarding the above-mentioned 8 domains, they should be repeated there to be clear if all of them are studied. There are no hypotheses formulated. This part must be developed/ improved.
Author’s reply: Aim was revised: this study aims to evaluate the baseline characteristics of these 8 domains, namely housing, outdoor spaces and buildings; transportation; job opportunities and civic participation; social participation; respect and social inclusion; communication and information; and community support and health services in Ipoh city….
- Generally, the presentation of section Material and Methodsshould be divided into subsections to present clearly the Material, e.g. sample selection and number of respondents divided into age, how many of them represent the city centre and non-city centre of Ipoh, also the form / scope of questionnaire should be described – due to many important aspects asked, they should be presented in a table, etc.
Author’s reply: Sample were conveniently sampled but not stratified according to age and city-centre or non-city centre. We understood that this may cause underpower, and hence added this into the limitation. The scope of the questionnaire was presented under the 3rd and 6th paragraph of the Materials and method, further supplemented with the full version of the questionnaire.
The used Methods are not clear – they need to be explained in a more scientific way, more professional, also they must be clearly referred to the sources to confirmed them as typical or not for that kind of study and argued.
Author’s reply: The study design is cross-sectional, which is typical and appropriate to answer research question of such nature. Authors has added further information to outline the strength of this study design, supported with references of studies with similar context. Text added: Cross-sectional study design was employed, as it is typical for population-based surveys, allowing estimation of prevalence and establishing association, which is appropriate for prompt public health planning, assessment and informing policies
The present form of this section is difficult to understand, especially the stages of the study and their role for possible results.
Author’s reply: Methods part was revised, with sub-headings to guide the readers for stages of study, including study design, study population, development of study instrument, data collection and data analysis.
- In the section of Results, the characteristics of respondents could be a bit developed to highlights more its main specific in the context of place of living or status; the direction in this case should follow then the order of presentation of differences in the further presentation of results.
Author’s reply: We highlight the place of living and living status in the first paragraph of results. We also inserted sub-headings for the results section, to guide the order of presentation. The flow of presentation was also in accordance to the Table to ease reading, started with 1st paragraph on demographic characteristics, the differences between city centre and non-city centre, 2nd paragraph on the overall perceived gap scores, 3rd paragraph on the sub-analyses of perceived gap scores between city centre and non-city centre population, and the subsequent paragraphs on Multivariate binary logistic regressions
The description of results itself is not much developed. The division into 8 domains is correct, however the high number of diversity between different types of respondents make it a bit difficult to notice.
Author’s reply: We agreed that there is a high diversity in term of perceived gaps among city-centre and non-city centre population, and we have revised this in the 3rd paragraph of the results. The results percentage of each variable was demonstrated, together with its domain, and we added p-value for example “a range of volunteer activities (29.9% vs 22.2%, D6, p=0.009)”. We also bolded the entire row of gap scores with significant p-values in Table 2, to make it easier for readers to notice the variables with significant differences.
The organization/presentation of Table 3 is weak, it must be improved, e.g. divided into separate or some more individual sections, to present the completed data better.
Author’s reply: We agreed with the reviewer comments. Hence, we have converted the Table 3 into Figure 2, and presented as individual sections which are clearer to view.
There is a lack of description/presentation in the text related to the significance of obtained results, for example in the relation to demographic characteristics of respondents, etc., presented in Table 3. That aspects also needs interpretation.
Author’s reply: The odds ratio of respective significant variables in Figure 2 were interpreted and added into the text.
- Despite some gaps in most sections, Discussion is quite strong part of the manuscript. However, the description is too long and due to the presentation of many (too many?) aspects in a very descriptive way, it becomes distracting and it from is a kind of a long “story” - it is difficult to find out what are the most valuable results of the study? The Authors should present the discussed aspects more synthetically and focus on their specifics /innovation - to present clearly the basic gaps of age-friendliness among community-dwelling older adults, and relate them directly to other studies.
Author’s reply: We agreed that it is important for Discussion to be succinct. Hence, it was cut-short 200 words from 2300 to 2100 words. We admit that it is a challenge to make the discussion concise, but at the same time comprehensive for the readers. The basic gaps of age-friendliness among community-dwelling older adults were depicted in the following sequence: housing, transportation, parking difficulties, home visit service, specialized care and nursing homes, outdoor facilities, job oppurtunities, social participation, civic engagement, communication and information, internet access, encompassing all eight domains of the age-friendly city requirements. We have made relevant citations and compared directly with other studies whenever it was applicable, including studies from UK, Europe, America. The most valuable results of the study often appears at the last sentence of each Discussion paragraph, which the authors tried to make practical and innovative recommendations where future works may be built further upon what was recommended, in the context of an age-friendly city.
- Conclusions are very short, and rather repeat general information from the Discussion, than summarize the manuscript and findings.
It must be clearly defined what the conducted research brings, especially in scientific (and practical?) context. Also the main direction in which interventions should be taken in all 8 domains must be proposed. In its present form, the conclusions are insufficient and not much linked to the research findings and their discussion.
Author’s reply: Conclusion was revised as such: Out of the eight domains, the most notable unmet needs were observed in the housing, transportation and job opportunities domains. Distinct differences in perceived gap was also observed between older population in the city-centre and non-city centre in six out of the eight domains. Concerted municipal strategies, political commitment and benchmarking of established age-friendly cities are warranted to address specific gaps in the local age-friendly context.
- Others:
- Figure 1 has a poor resolution and is not legible - it needs improvement.
Author’s reply: We have improved the resolution for Figure 1 by uploading a new version
- there are some punctuation errors in the text - lack of dots, etc.
Author’s reply: We have rechecked the punctuations.
- the way of writing references in text should be in square bracket
Author’s reply: We have change the references in text into square bracket.
Reviewer 2 Report
Comments
Housing manifests the cultural diversity of the various peoples and a collection of their daily practices. From generation to generation people make their life journey where there are moments of reaction and adaptation in accordance to their biological functioning. Knowing that predictability is fortuitous and fallible, and that statistics are rarely assertive, reality becomes self-explanatory. In regards to the elderly, that is, above 65/80 year-olds, we can have these people live in a dignified and balanced way. The key observation in that housing services are not a marginal component of consumption.
Good housing achievements also require consideration of age, magnitude, healthcare, complexity, probability, duration, job location, frequency and reversibility of the risk, also taking into account the vulnerability and resilience of the values exposed to risk. This happen in Ipoh as well. Without any concern for inclusiveness, comes only from above the arguments that demonstrate the need to reorient social protection - effective and fair - in identifying social vulnerabilities, in order to develop self-protection capabilities and reinforce resilience.
This situation, which will continue for many years, will lead to generational issues regarding the change to a new and desirable social model for the EU. The dynamics of social relations and, in this case, housing relations, results from the way a group of people have acted based on their habits, customs and experiences. Those over 65, or even those over 80, are different from other citizens in educational and professional terms.
Sugestions
Risk volatility should be highlighted in the article or in future research work on this topic.
As a suggestion, the State must adopt preventative measures in the field of social protection, even if this implies a tax increase. We must consider not only public capacity, but also existing private capabilities. On the one hand, we must know not only the individual needs of social protection, but also the individual or collective capacities for self-protection and the resilience of individuals and age groups in each city. This way, social inclusion is more just because it is aimed essentially at the most deprived, those who cannot take self-protection measures or recover after any financial or economic crisis in old age. The crucial risk and fear for the elderly is financial and economic.
On the other hand, designing social inclusion policies in order to protect those who need them, in addition to being fairer, is more effective, because although the value of human life is absolute, better protection of the most vulnerable makes it possible to maximize existing capacities in society without duplicating efforts, thus obtaining better results in terms of effectiveness in social anxiety. In short, the advantage of identifying vulnerabilities is that this creates synergies between State social protection and social policies, establishing new intervention priorities and reorienting social protection to a scale of values.
Another point to note as a suggestion for research is that old habits take a long time to disappear, and the usual skills of politicians are more directed to exercising authority than to regulation. And these are difficult habits to overcome when the State has the most resources, which are under the direct control of political leaders.
So, face-to-face or telephone interviews should worry and look to the future, even if it is not based in a Malaysian city. Housing affordability is a major concern for older people, especially vulnerable individuals with low income, especially for the oldest.
Author Response
Risk volatility should be highlighted in the article or in future research work on this topic.
Author’s reply: Revised discussion: The lack of employment prospects may result in financial and economic volatility.
As a suggestion, the State must adopt preventative measures in the field of social protection, even if this implies a tax increase. We must consider not only public capacity, but also existing private capabilities. On the one hand, we must know not only the individual needs of social protection, but also the individual or collective capacities for self-protection and the resilience of individuals and age groups in each city. This way, social inclusion is more just because it is aimed essentially at the most deprived, those who cannot take self-protection measures or recover after any financial or economic crisis in old age. The crucial risk and fear for the elderly is financial and economic.
Author’s reply: Revised discussion:Hence, the government should take precautionary actions in the realm of social protection, through revision of tax system and tapping into existing private capacity to complement current public resources (Shaikh et al. 2011). Meanwhile, individuals’ demands for social safety, self-protection skills and resilience should be understood and addressed through social inclusion policies.
On the other hand, designing social inclusion policies in order to protect those who need them, in addition to being fairer, is more effective, because although the value of human life is absolute, better protection of the most vulnerable makes it possible to maximize existing capacities in society without duplicating efforts, thus obtaining better results in terms of effectiveness in social anxiety. In short, the advantage of identifying vulnerabilities is that this creates synergies between State social protection and social policies, establishing new intervention priorities and reorienting social protection to a scale of values.
Author’s reply: Revised discussion:Better protection of the most vulnerable allows society to maximise existing capacities without duplicating efforts, resulting in better alleviation of social anxiety. Recognising vulnerabilities produces synergies between state social protection and social policies, lead to innovative interventions and reorienting social protection (Adato & Bassett, 2009).
Another point to note as a suggestion for research is that old habits take a long time to disappear, and the usual skills of politicians are more directed to exercising authority than to regulation. And these are difficult habits to overcome when the State has the most resources, which are under the direct control of political leaders.
Author’s reply: Revised discussion: While the State has the largest resources, a paradigm shift is difficult, as administrators are geared towards bureaucracy than implementing new policies.
So, face-to-face or telephone interviews should worry and look to the future, even if it is not based in a Malaysian city.
Author’s reply: Revised discussion:Studies with comparable methodology may be replicated in other countries to validate the findings in future.
Housing affordability is a major concern for older people, especially vulnerable individuals with low income, especially for the oldest.
Author’s reply: Revised discussion:This was added in the second paragraph of discussion: Housing affordability is a major concern for older people, especially vulnerable individuals with low income
Reviewer 3 Report
I have now read this revised manuscript and find it relevant. Nevertheless, I would suggest some minor changes before publishing. I find missing a mention of the WHO, as a promoter of the Age-Friendly Cities project, as well as the paradigm of active aging. In the same way, I find missing what exactly it means for a city to become part of the World Network of Age-Friendly Cities. There is abundant bibliography on the matter.
Author Response
I have now read this revised manuscript and find it relevant. Nevertheless, I would suggest some minor changes before publishing. I find missing a mention of the WHO, as a promoter of the Age-Friendly Cities project, as well as the paradigm of active aging.
Author’s reply: Revised and added in Introduction: According to the WHO initiative, an "age-friendly" city is one that encourages active aging; it maximises possibilities for health, involvement, and security in order to improve people's quality of life as they age (Plouffe et al. 2010).
In the same way, I find missing what exactly it means for a city to become part of the World Network of Age-Friendly Cities. There is abundant bibliography on the matter.
Author’s reply: Revised and added in Introduction: Membership in the Network does not imply age-friendliness. Rather, it represents cities' dedication to listening the needs of ageing populations, assessing and monitoring age-friendliness features, and collaborating with older people and across sectors to develop age-friendly physical and social settings, and the commitment to share one's experience, accomplishments, and lessons learned with other cities and communities (Flores et al. 2019).
Round 2
Reviewer 1 Report
Manuscript ID: ijerph-1694119
Title: Perceived gap of age-friendliness among community-dwelling older adults: Findings from a middle-income country
Authors: Chee Tao Chang*, Xin Jie Lim, Premaa Supramaniam, Chii Chii Chew, Lay Ming Ding, Philip Rajan
I appreciate all corrections made by Authors, most of suggested corrections has been applied, even if the scope of made changes/improvement is not very deep. They generally increase the quality of most sections of the manuscript.
- the Abstract has been improved;
- the Introduction section has been developed, however, the number of cited literature items is still quite limited; the main aim is shortly but better presented;
- the presentation of Material and Methods, including its organization, is more clear, also the description of used method is more clear;
- the section of Results has been improved, generally by adding some missed data; also Table 2 is much better organized and, thus, clear;
- the aspects in section of Discussion are well described.
My last suggestions:
- Table 2 has clear order and division, however, the symbols of each Domain, such as D1, D2, …, are used in the text, but they must also be added to Table 2, otherwise this relation is very difficult to find out;
- Conclusions are still very short – 3 sentences only while Discussion is very rich (?). Authors use a kind of 'mental shortcuts' again, and did not even repeat the main information that the results relate to Ipoh in Malaysia... Especially the last sentence is very general and can be used for many similar papers/research. Conclusions still need to be developed in order to inform what the study brings - e.g. some most important recommendations for the authorities, etc.
- English corrections are still needed in my opinion to increase the quality of the manuscript.
- there are still some punctuation errors in the text - missed spaces before brackets, e.g. line 38,42, 48, and so on…
Author Response
- Table 2 has clear order and division, however, the symbols of each Domain, such as D1, D2, …, are used in the text, but they must also be added to Table 2, otherwise this relation is very difficult to find out;
Response 1: The symbols are added into Table 2.
- Conclusions are still very short – 3 sentences only while Discussion is very rich (?). Authors use a kind of 'mental shortcuts' again, and did not even repeat the main information that the results relate to Ipoh in Malaysia... Especially the last sentence is very general and can be used for many similar papers/research. Conclusions still need to be developed in order to inform what the study brings - e.g. some most important recommendations for the authorities, etc.
Response 2: The conclusion was developed further to include important recommendations for the authorities.
- English corrections are still needed in my opinion to increase the quality of the manuscript.
Response 3: The paper was sent to professional proof-reading services for improvement
- there are still some punctuation errors in the text - missed spaces before brackets, e.g. line 38,42, 48, and so on…
Response 4: The paper was sent to professional proof-reading services for improvement, and the punctuation errors were corrected.